# Dose-Dependent Proton Pump Inhibitor Exposure and Risk of Type 2 Diabetes: A Nationwide Nested Case–Control Study

**DOI:** 10.3390/ijerph19148739

**Published:** 2022-07-18

**Authors:** Hsin-Ya Kuo, Chih-Sung Liang, Shih-Jen Tsai, Tzeng-Ji Chen, Che-Sheng Chu, Mu-Hong Chen

**Affiliations:** 1Department of Psychiatry, Kaohsiung Veterans General Hospital, Kaohsiung City 813, Taiwan; hykuo@vghks.gov.tw; 2Department of Psychiatry, Tri-Service General Hospital, Beitou Branch, National Defense Medical Center, Taipei 112, Taiwan; lcsyfw@gmail.com; 3Department of Psychiatry, Taipei Veterans General Hospital, Taipei 112, Taiwan; tsai610913@gmail.com; 4Department of Psychiatry, College of Medicine, National Yang Ming Chiao Tung University, Taipei 112, Taiwan; 5Department of Family Medicine, Taipei Veterans General Hospital, Taipei 112, Taiwan; tjchen@vhct.gov.tw; 6Institute of Hospital and Health Care Administration, National Yang Ming Chiao Tung University, Taipei 112, Taiwan; 7Department of Family Medicine, Taipei Veterans General Hospital, Hsinchu Branch, Hsinchu 31064, Taiwan; 8Center for Geriatrics and Gerontology, Kaohsiung Veterans General Hospital, Kaohsiung City 813, Taiwan; 9Non-Invasive Neuromodulation Consortium for Mental Disorders, Society of Psychophysiology, Taipei 114, Taiwan; 10Graduate Institute of Medicine, College of Medicine, Kaohsiung Medical University, Kaohsiung City 80708, Taiwan

**Keywords:** proton pump inhibitors, type 2 diabetes mellitus, upper gastrointestinal disease

## Abstract

Background: To investigate the association between proton pump inhibitor (PPI) exposure and a risk of type 2 diabetes mellitus (T2DM) among patients with upper gastrointestinal disease (UGID). Method: We conducted a case–control study from Taiwan’s National Health Insurance Research Database between 1998 and 2013. A total of 20,940 patients with T2DM and 20,940 controls were included. The dose of PPIs was categorized according to the cumulative defined daily dose (cDDD). The risk of T2DM was assessed using conditional logistic regression analysis. Result: Compared with cDDD ≤ 30, higher dosage of PPI exposure was associated with an increased risk of T2DM development: cDDD 31–120 (odds ratio [OR]: 1.20, 95% confidence interval [CI]: 1.13–1.26); cDDD 121–365 (OR: 1.26, 95% CI: 1.19–1.33); and cDDD > 365 (OR: 1.34, 95% CI: 1.23–1.46). Subgroup analysis of individual PPI showed that pantoprazole (OR: 1.14, 95% CI: 1.07–1.21), lansoprazole (OR: 1.08, 95% CI: 1.03–1.12), and omeprazole (OR: 1.11, 95% CI: 1.06–1.16) have a significantly higher risk of T2DM development. Conclusions: A dose-dependent increased risk of T2DM was found among patients with UGID using higher doses of PPIs compared with those with lower doses of these drugs. Further studies are necessary to investigate the underlying pathophysiology of PPIs and T2DM.

## 1. Introduction

Proton pump inhibitors (PPIs) are commonly prescribed to treat peptic acid diseases. To date, PPIs have been among the most frequently prescribed medications [1]; however, PPIs also have problems with overuse or misuse without clear and appropriate indications. Some prescription guidelines recommended 4–8 weeks of treatment with empirical PPIs for associated diseases [2,3]. However, inappropriate PPI prescriptions and misuse have been documented in numerous studies [4,5]. A long-term use of PPIs has been suggested to increase a risk of gastrointestinal infection [6,7], bone diseases [8], and malabsorption of vitamins [9]. An association between PPI exposure and type 2 diabetes mellitus (T2DM) has also been reported [10,11].

Diabetes is a major global health concern. In 2019, it was estimated that 463 million people were living with diabetes [12]. Patients with T2DM are more likely to develop hypertension, obesity, hyperlipidemia, and cardiovascular disease (CVD) [13,14]. In addition, diabetic patients are associated with a unique and widespread form of atherosclerosis and increased risk of restenosis after revascularization for CVD [15,16], resulting in CVD itself as a major cause of mortality and morbidities in this population [17]. In clinical practice, it is common for patients with CVD receiving PPIs to prevent gastrointestinal tract bleeding in the treatment with anticoagulants and antiplatelets. Taken together, it is vital to explore the potential association between the use of PPIs and development of T2DM. Furthermore, some studies suggest that PPI use may affect the gut microbiota composition and impact its health, which may lead to gut inflammation and changes to the host’s metabolism [18]. The gut microbiota has also been suggested to play an important role in the onset and progression of T2DM [19].

The association between long-term PPI use and the risk of T2DM is controversial. A prospective analysis showed that the long-term use of PPIs was associated with a 24% increased risk of diabetes [11], although they focused on health professionals; thus, the findings could not be applied to the general population. In addition, the study did not provide details of PPI dosage, frequency, brand, or indications. A prospective population-based cohort study showed that PPI use was associated with a 1.7-times increased risk of incident T2DM [10]. However, contrary results of no association between PPIs and T2DM have been reported in other studies [20,21], including the weakness of not comparing the difference between individual PPI and the follow-up period, which was relatively short.

Given the limitations of the above studies, we conducted the present study to investigate the association between PPI use and T2DM development among a community-dwelling population in Taiwan, based on the National Health Insurance Research Database (NHIRD). In addition, we demonstrated whether individual PPI might contribute to the different risks of T2DM and explored the possible dose-dependent effects.

## 2. Materials and Methods

### 2.1. Data Source

This nested case–control study used data from the Taiwan NHIRD from January 1998 to December 2013. The details of the NHIRD have been described in our previous research [22,23,24,25]. Briefly, the National Health Research Institute audits and releases the NHIRD for scientific and study purposes. The NHIRD has detailed healthcare information on 23 million enrollees (99.9% of the population of Taiwan) based on a random sample of all enrollees of the NHI program. There were no significant differences in age, sex, or healthcare costs between the sample group and the enrollees. The diagnostic codes used were based on the International Classification of Diseases, 9th Revision, Clinical Modification (ICD-9-CM). The study protocol was reviewed by the institutional review board of the Taipei Veterans General Hospital.

### 2.2. Inclusion Definition for Selection of Cases and Controls

Individuals exposed to any PPI (pantoprazole, lansoprazole, omeprazole, esomeprazole, and rabeprazole) for the corresponding indications (peptic ulcer, gastroesophageal reflux disorder, and upper gastrointestinal bleeding) were eligible for inclusion in this study. Individuals without a history of diabetes mellitus (ICD-9-CM code: 250) at the time of enrollment and those who subsequently developed T2DM (ICD-9-CM codes: ICD-9-CM code: 250.x0 and 250.x2, x = 0–9) diagnosed by board-certified pediatricians, internal medicine physicians, endocrinologists, and family medicine physicians based on laboratory examination during the follow-up period were enrolled as the case group. The diagnostic validity (positive predictive value = 0.92) of T2DM was established in the NHIRD [26]. The time of the first PPI exposure and the time of T2DM diagnosis were defined as the enrollment and endpoint time, respectively. The period between enrollment and the endpoint was the follow-up period. The matched (1:1) control group was randomly selected based on age, sex, residence, income, enrollment time, endpoint time, follow-up period, and medical comorbidities from those who were exposed to any PPI for the corresponding indications after eliminating the case group and those who had been diagnosed with diabetes mellitus at any time.

### 2.3. PPI Exposure

The defined daily dose (DDD) recommended by the World Health Organization (WHO) is a unit for measuring the prescribed number of drugs. The DDD is the assumed average maintenance dose per day of a drug consumed for its main indication. We calculated the sum of the dispensed DDD as the cumulative DDD (cDDD) of PPI use during the follow-up period. Based on the cDDD, PPI use patterns were separated into four subgroups: cDDD ≤ 30, cDDD 31–120, cDDD 121–365, and cDDD > 365. Such dosage intervals were based on the methodology of previous studies [27,28]. Patients in the cDDD > 365 group may represent a person who had continuously received PPI treatment at a standard dosage for one year. Using the cDDD, we enabled a comparison of drug usage between different drugs in the same group and allowed for the examination of the dose–response effect of PPIs on T2DM, which was commonly used in previous pharmacoepidemiologic studies of PPIs [29,30,31]. 

### 2.4. Covariates

Obesity, dyslipidemia, hypertension, and alcohol use disorders were assessed as confounding factors. In addition, the level of urbanization (levels 1–5; level 1, most urbanized region; level 5, least urbanized region) was examined.

### 2.5. Statistical Analysis

For between-group comparisons, the independent *t*-test was used for continuous variables and the Pearson’s X^2^ test was used for nominal variables when appropriate. Conditional logistic regression analysis was used to investigate the likelihood of subsequent T2DM after adjusting for demographic data (age, sex, residence, and income), indications of PPI use, and medical comorbidities (obesity, dyslipidemia, hypertension, and alcohol use disorders) among PPI users (cDDD categories: ≤30, 31–120, 121–365, and >365). Subgroup analysis stratified by sex and each PPI category (pantoprazole, lansoprazole, omeprazole, esomeprazole, and rabeprazole) was also performed. Sensitivity analysis, after excluding medical comorbidities (obesity, dyslipidemia, hypertension, and alcohol use disorders), was performed to examine the independent effect of PPI exposure on subsequent T2DM risk. A two-tailed *p*-value < 0.05 was considered statistically significant. All data processing and statistical analyses were performed using Statistical Package for Social Sciences (SPSS) version 17 software (SPSS Inc.) and Statistical Analysis Software (SAS) version 9.1 (SAS Institute, Cary, NC, USA). 

## 3. Results

### 3.1. Characteristics of Study Participants

Data from 41,880 patients who used PPIs (20,940 in the T2DM group and 20,940 in the non-T2DM group) were analyzed. The distribution of demographic characteristics, indication for PPI use, and cDDD category are presented in Table 1. The mean ages at enrollment/first time of PPI exposure in the T2DM and non-T2DM groups were 55.82 ± 13.48 and 55.87 ± 13.55 years, respectively. The T2DM group had a significantly higher mean cDDD than that of the non-T2DM group (140.78 ± 230.37 vs. 127.59 ± 199.38, *p* < 0.001). There was no significant difference in the indication of PPI use between the T2DM and non-T2DM groups. However, patients were mostly treated with PPIs under the indication of peptic ulcers in both groups (97.1% vs. 97.0% in the T2DM and non-T2DM groups, respectively). 

### 3.2. Associations between PPI Exposure and the Risk of T2DM 

After adjusting for demographic data, indications for PPI use, and medical comorbidities, the association between PPI exposure and T2DM is shown in Table 2. There was also dose-dependency in the association between PPI dosage and the increased risk of T2DM (cDDD 31–120, odds ratio [OR]: 1.20; 95% CI: 1.13–1.26; cDDD 121–365, OR: 1.26, 95% CI: 1.19–1.33; cDDD > 365, OR: 1.34, 95% CI: 1.23–1.46) compared with PPI dosage of cDDD ≤30. Furthermore, the dose-dependent associations persisted when the patients were divided into male and female groups. 

After excluding medical comorbidities as covariates, sensitivity analysis revealed similar results (Table 3). In addition, the association between PPI exposure and T2DM increased with a higher dosage of PPI use (cDDD 31–120, OR: 1.12, 95% CI: 1.05–1.20; cDDD 121–365, OR: 1.20, 95% CI: 1.12–1.29; and cDDD > 365, OR: 1.40, 95% CI: 1.35–1.57) compared with a PPI dosage of cDDD ≤ 30.

### 3.3. Associations between Individual PPI and the Risk of T2DM

When divided into individual PPI, pantoprazole (OR: 1.14, 95% CI: 1.07–1.21), lansoprazole (OR: 1.08, 95% CI: 1.03–1.12), and omeprazole (OR: 1.11, 95% CI: 1.06–1.16) were found to be associated with T2DM development (Table 4; all *p* < 0.05), whereas esomeprazole (OR: 1.05, 95% CI: 1.00–1.10) or rabeprazole (OR: 0.96, 95% CI: 0.89–1.04) did not reveal significant differences.

## 4. Discussion

In the present study, our main findings showed the following: (a) a dose-dependent effect was found between PPI use and the increased risk of T2DM; (b) subgroup analysis showed that the risk of T2DM significantly increased in both male and female patients receiving PPIs; (c) sensitivity analysis of the exclusion of medical comorbidities showed similar findings of increased risk of T2DM in patients receiving PPI treatment; and (d) when stratified by individual PPI, the risk of T2DM was increased among patients receiving pantoprazole, lansoprazole, and omeprazole, but not esomeprazole or rabeprazole.

### 4.1. Comparison of Other Studies

Our results show that PPI exposure may increase T2DM risk in a dose-dependent manner. Several studies have also reported similar results. A large randomized controlled study conducted by 17,598 participants investigated the safety of pantoprazole and found a trend of association between pantoprazole exposure and the risk of diabetes, although not significantly (OR: 1.15, 95% CI: 0.89–1.50) [20].

A recent prospective cohort study conducted by Yuan et al. recruited 204,689 health professionals and showed that regular PPI use was associated with a 24% higher risk of T2DM (hazard ratio [HR] 1.24, 95% CI: 1.17–1.31), and the risk increased with a longer duration of PPI exposure [11]. However, this study was based on a certain population, rather than the general population, and did not provide detailed information regarding the dosage, brand, frequency, or indication of PPI use. Another prospective cohort study conducted with 9531 participants showed that incident PPI use was associated with a significantly increased risk of T2DM, with a hazard of 1.69 (95% CI: 1.36–2.10) [10]. In addition, they found that the risk of T2DM increased with a higher dosage and longer duration of PPI exposure, which is consistent with our findings.

On the other hand, other studies have shown contrary results to our study. For example, a retrospective cohort study by Lin et al. based on the NHIRD included 388,098 patients, which showed that patients with upper gastrointestinal disease (UGID) receiving PPIs had a 20% decreased risk of diabetes compared with patients with UGID without PPI use over a 5-year follow-up period (HR: 0.80, 95% CI: 0.73–0.88) [21]. This difference may be attributed to different methodological designs and populations. Compared with Lin et al., the present study has several advantages, including a longer follow-up period and further matched control samples by controlling residence, income, enrollment time, endpoint time, follow-up period, and medical comorbidities.

Few studies have investigated the differences between individual brands of PPI. Our analysis of different PPI showed an association between T2DM and pantoprazole, lansoprazole, and omeprazole. This finding is similar to that of Czarniak et al. [10], who found a higher risk of T2DM with pantoprazole, omeprazole, and esomeprazole. The potency of the five PPIs, from low to high, was pantoprazole, lansoprazole, omeprazole, esomeprazole, and rabeprazole [32]. Our findings showed that PPIs with lower potency seem to have a higher risk of T2DM. Patients taking PPIs with relatively lower potency, such as pantoprazole, may need higher equivalent dosage to reach the anti-acid effect compared with those with relatively higher potency. For example, 20 mg of esomeprazole is approximately equivalent to 120 mg of pantoprazole [33]. Higher dosage exposure to PPIs with relatively lower potency may partially explain the increased risk of T2DM. Further studies investigating the underlying mechanism of PPIs and their metabolic effects are needed.

### 4.2. Possible Mechanism

The mechanism underlying the association between PPI exposure and T2DM remains uncertain. A recent study investigating PPI use and coronavirus disease 2019 (COVID-19) severity found that PPI exposure was associated with elevated fasting blood glucose levels [34]. Some studies have suggested that PPI usage may decrease Shannon’s diversity and impact the health of the gut microbiota [35,36]. The gut microbiota has been suggested to play an important role in the onset and progression of T2DM [19]. In addition, PPI usage is associated with hypomagnesemia [37], and some studies have demonstrated a link between hypomagnesemia and T2DM by altering cellular glucose transport [38]. However, the interaction between PPI exposure and T2DM may be complex. Patients diagnosed with T2DM are more likely to report gastroenterological complaints [39] and may be associated with higher levels of anti-acid medication exposure.

### 4.3. Strengths and Limitations 

The present study had several noteworthy strengths. First, we conducted a nationwide population-based study with a large, unbiased, and well-defined patient sample; therefore, our study minimized the selection bias. Second, our study focused on analyzing data from patients with UGID instead of data from the general population, which may decrease the indication bias. Third, we matched the control group with multiple covariates, such as residence, income, enrollment time, endpoint time, follow-up period, and medical comorbidities, to decrease potential confounding factors. Fourth, the present study further investigated the differences between individual PPI and the risk of T2DM.

Nevertheless, our study had several limitations. First, this was an observational study from the NHIRD; therefore, it was impossible to infer causality from the results. In addition, a family history of diabetes mellitus, lifestyle factors (such as diet, physical activity, and alcohol consumption), and environmental factors (such as air pollution and cigarette exposure) were not documented in the NHIRD and were not controlled for or addressed in the present study. Second, we were unable to assess medical adherence to medications; therefore, in patients with poor compliance, the risk of T2DM may have been underestimated. Third, as the diagnoses were identified using ICD codes, the risk of T2DM development may have been underestimated because only patients seeking medical help were included in this study. Finally, we could not exclude patients whose T2DM was diagnosed before the establishment of the NHIRD and those who were undiagnosed before the index date.

## 5. Conclusions

Our results showed a dose-dependent increased risk of T2DM among patients with UGID receiving PPIs compared with those without PPIs. Further studies are required to explore the possible pathophysiological mechanisms underlying this association.

## Figures and Tables

**Table 1 ijerph-19-08739-t001:** Demographic characteristics of patients who were exposed to PPIs with or without subsequent type 2 diabetes.

	Patients Exposed to PPIs		*p*-Value
	With Subsequent Type 2 Diabetes (*n* = 20,940)	Without Subsequent Type 2 Diabetes (*n* = 20,940)	
Age at enrollment/first time of PPI exposure (years, SD)	55.82 (13.48)	55.87 (13.55)	0.702
Sex (*n*, %)			1.000
Male	12,032 (57.5)	12,032 (57.5)	
Female	8908 (42.5)	8908 (42.5)	
Indications of PPI use (*n*, %)			
Peptic ulcer	20,336 (97.1)	20,319 (97.0)	0.622
Gastroesophageal reflux disorder	2840 (13.6)	2788 (13.3)	0.465
Upper gastrointestinal bleeding	5383 (25.7)	5543 (26.5)	0.077
Medical comorbidities at enrollment (*n*, %)			
Obesity	158 (0.8)	158 (0.8)	1.000
Hypertension	6894 (32.9)	6894 (32.9)	1.000
Dyslipidemia	3108 (14.9)	3108 (14.9)	1.000
Alcohol use disorders	1038 (5.0)	1038 (5.0)	1.000
Use of PPIs during follow-up period (*n*, %)			<0.001
cDDD, >365	1478 (7.1)	1298 (6.2)	
cDDD, 121~365	5936 (28.3)	5534 (26.4)	
cDDD, 31~120	8771 (41.9)	8552 (40.9)	
cDDD, ≤30	4755 (22.7)	5556 (26.5)	
Mean cDDD (SD)	140.78 (230.37)	127.59 (199.38)	<0.001
Level of urbanization (*n*, %)			1.000
1 (most urbanized)	2483 (11.9)	2483 (11.9)	
2	5076 (24.2)	5076 (24.2)	
3	1760 (8.4)	1760 (8.4)	
4	2042 (9.8)	2042 (9.8)	
5 (most rural)	9579 (45.7)	9579 (45.7)	
Income-related insured amount			1.000
TWD ≤15,840/mo	8139 (38.9)	8139 (38.9)	
TWD 15,841–25,000/mo	8377 (40.0)	8377 (40.0)	
TWD ≥25,001/mo	4424 (21.1)	4424 (21.1)	
Follow-up duration (years, SD)	4.42 (3.41)	4.37 (3.43)	0.131

**Table 2 ijerph-19-08739-t002:** Logistical regression models of risk of type 2 diabetes among patients who were exposed to PPIs.

	Males	Females	All
	OR ^a^ (95% CI)	OR ^a^ (95% CI)	OR ^a^ (95% CI)
Use of PPIs during follow-up period			
cDDD, ≤30	1 (reference)	1 (reference)	1 (reference)
cDDD, 31~120	**1.21 (1.13–1.29)**	**1.18 (1.10–1.27)**	**1.20 (1.13–1.26)**
cDDD, 121~365	**1.26 (1.18–1.36)**	**1.25 (1.15–1.35)**	**1.26 (1.19–1.33)**
cDDD, >365	**1.33 (1.19–1.48)**	**1.37 (1.20–1.56)**	**1.34 (1.23–1.46)**

PPIs: proton pump inhibitor; cDDD: cumulative defined daily dose; OR: odds ratio; CI: confidence interval. ^a^ adjusted for demographic data, indications of PPI use, and medical comorbidities. Bold type indicates statistical significance.

**Table 3 ijerph-19-08739-t003:** Sensitivity test of risk of type 2 diabetes among patients who were exposed to PPIs.

	Total	Excluding Medical Comorbidities
	OR ^a^ (95% CI)	OR ^b^ (95% CI)
Use of PPIs during follow-up period		
cDDD, ≤30	1 (reference)	1 (reference)
cDDD, 31~120	**1.20 (1.13–1.26)**	**1.12 (1.05–1.20)**
cDDD, 121~365	**1.26 (1.19–1.33)**	**1.20 (1.12–1.29)**
cDDD, >365	**1.34 (1.23–1.46)**	**1.40 (1.35–1.57)**

PPIs: proton pump inhibitor; cDDD: cumulative defined daily dose; OR: odds ratio; CI: confidence interval. ^a^ adjusted for demographic data, indications of PPI use, and medical comorbidities. ^b^ adjusted for demographic data and indications of PPI use. Bold type indicates statistical significance.

**Table 4 ijerph-19-08739-t004:** Risk of type 2 diabetes among patients who were exposed to PPIs, stratified by each PPI.

	With Subsequent Type 2 Diabetes (*n*, %)	Without Subsequent Type 2 Diabetes (*n*, %)	OR ^a^ (95% CI)
Pantoprazole	2472 (11.8)	2206 (10.5)	1.14 (1.07–1.21)
Lansoprazole	5697 (27.2)	5407 (25.8)	1.08 (1.03–1.12)
Omeprazole	5015 (23.9)	4658 (22.2)	1.11 (1.06–1.16)
Esomeprazole	3858 (18.4)	3808 (17.7)	1.05 (1.00–1.10)
Rabeprazole	1364 (6.5)	1411 (6.7)	0.96 (0.89–1.04)

PPIs: proton pump inhibitor; OR: odds ratio; CI: confidence interval. ^a^ adjusted for demographic data, indications of PPI use, Charlson Comorbidity Index score, and medical and psychiatric comorbidities. Bold type indicates statistical significance.

## Data Availability

The data that support the findings of the study are available from the corresponding author upon reasonable request.

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
