# Peer review of "Dose-Dependent Proton Pump Inhibitor Exposure and Risk of Type 2 Diabetes: A Nationwide Nested Case–Control Study"

_ijerph, 2022, doi:10.3390/ijerph19148739_

Round 1
Reviewer 1 Report
The article written by Kuo et al. is related to a very important issue such as the risk of diabetes mellitus developing associated with the use of proton pump inhibitors (PPIs). PPIs are one of the most often overprescribed drugs in clinical practice. Although generally well-tolerated, they are not without the risk of complications. On the other hand, diabetes mellitus forms one of the most crucial problems for public health worldwide.
Although as told above the paper is generally well prepared and should be considered for publication in the International Journal of Environmental Research and Public Health, some revisions are necessary, which may further improve the quality of the manuscript.
Major revisions:
1) Although the Authors wrote sufficient information about the PPIs in the Introduction, information about diabetes mellitus is laconic and should be improved. The Authors wrote only information about the estimated number of people with diabetes worldwide in 2019. In my opinion, it should be elucidated why diabetes mellitus is so important problem. First of all, it should be mentioned that diabetes mellitus is a very strong risk factor for developing cardiovascular diseases, and on the other hand cardiovascular diseases are the most common cause of mortality and morbidity in the population of patients living with diabetes. Diabetes mellitus is associated with accelerated and changed the course of the development of atherosclerosis and increased risk of restenosis after endovascular treatment of atherosclerotic cardiovascular diseases (see for example the following recently published references: 10.3390/antiox11050856; 10.3390/ijerph182211970; 10.1007/s11886-019-1107-y; 10.1177/2047487319878371; 10.4239/wjd.v6.i7.961). Moreover, in clinical practice PPIs are commonly prescribed in patients with cardiovascular diseases to prevent gastrointestinal tract bleeding in the course of the treatment with anticoagulants and antiplatelets which makes this potential association (between PPIs and diabetes) even more important.
2) I do not understand the concept of cDDD. The Authors wrote that cDDD is the sum of DDD for all the PPIs. But one patient takes only one PPI at the moment. So why do you use the sum for all PPIs? It is not clear for me, please explain.
3) In the Discussion, when you cite the results obtained by other researchers and use hazard or odds ratio, give also always confidence interval, please (for example line 207).
4) In the Discussion the Authors used such expression as low-potency and high-potency PPIs. Please, explain precisely which substances are low-patency PPIs and which ones are high-patency.
5) Please, comment how dosage intervals used in the analysis are related to typical doses of PPIs used in the clinical practice.
Minor revisions:
1) In the main text references should be marked in square brackets, e.g. [1], [2,3], etc.
2) Please check the work very carefully for minor editing errors, such as the lack of spaces between some words or sentences, as in lines 71, 74, 96, 200, 205.
3) List of references should be written in accordance with the rules of MDPI.
Author Response
#Reviewer 1
The article written by Kuo et al. is related to a very important issue such as the risk of diabetes mellitus developing associated with the use of proton pump inhibitors (PPIs). PPIs are one of the most often overprescribed drugs in clinical practice. Although generally well-tolerated, they are not without the risk of complications. On the other hand, diabetes mellitus forms one of the most crucial problems for public health worldwide. Although as told above the paper is generally well prepared and should be considered for publication in the International Journal of Environmental Research and Public Health, some revisions are necessary, which may further improve the quality of the manuscript.
Response to Comment:
We appreciate the reviewer’s time and effort in enhancing the quality of our work. We have amended the manuscript following your suggestions. We hope that we have appropriately addressed all the comments and the manuscript is now suitable for publication.
Comment-1
Although the Authors wrote sufficient information about the PPIs in the Introduction, information about diabetes mellitus is laconic and should be improved. The Authors wrote only information about the estimated number of people with diabetes worldwide in 2019. In my opinion, it should be elucidated why diabetes mellitus is so important problem. First of all, it should be mentioned that diabetes mellitus is a very strong risk factor for developing cardiovascular diseases, and on the other hand cardiovascular diseases are the most common cause of mortality and morbidity in the population of patients living with diabetes. Diabetes mellitus is associated with accelerated and changed the course of the development of atherosclerosis and increased risk of restenosis after endovascular treatment of atherosclerotic cardiovascular diseases (see for example the following recently published references: 10.3390/antiox11050856; 10.3390/ijerph182211970; 10.1007/s11886-019-1107-y; 10.1177/2047487319878371; 10.4239/wjd.v6.i7.961). Moreover, in clinical practice PPIs are commonly prescribed in patients with cardiovascular diseases to prevent gastrointestinal tract bleeding in the course of the treatment with anticoagulants and antiplatelets which makes this potential association (between PPIs and diabetes) even more important.
Response to Comment-1:
We are grateful for the very wisdom suggestion to improve this manuscript.
Diabetes is a major global health concern. In 2019, it was estimated that 463 million people were living with diabetes in 2019. Patients with T2DM were more likely to develop hypertension, obesity, hyperlipidemia, and cardiovascular disease (CVD) [Jakubiak GK et al., 2022; Glovaci et al., 2019]. In addition, diabetic patients were associated with unique and widespread form of atherosclerosis and increased risk of restenosis after revascularization for CVD [Jakubiak GK et al., 2021; Thiruvoipati T et al., 2015], resulting in CVD itself as a major cause of mortality and morbidities in this population [Dal Canto E et al., 2019]. In clinical practice, it is common for patients with CVD receiving PPIs to prevent gastrointestinal tract bleeding in the treatment with anticoagulants and antiplatelets. Taken together, it is vital to explore the potential association between the use of PPIs and development of T2DM.
New References
Dal Canto E, Ceriello A, Rydén L, Ferrini M, Hansen TB, Schnell O, Standl E, Beulens JW. Diabetes as a cardiovascular risk factor: An overview of global trends of macro and micro vascular complications. Eur J Prev Cardiol. 2019 Dec;26(2_suppl):25-32.
Glovaci D, Fan W, Wong ND. Epidemiology of Diabetes Mellitus and Cardiovascular Disease. Curr Cardiol Rep. 2019 Mar 4;21(4):21.
Jakubiak GK, Cieślar G, Stanek A. Nitrotyrosine, Nitrated Lipoproteins, and Cardiovascular Dysfunction in Patients with Type 2 Diabetes: What Do We Know and What Remains to Be Explained? Antioxidants (Basel). 2022 Apr 27;11(5):856.
Jakubiak GK, Pawlas N, Cieślar G, Stanek A. Pathogenesis and Clinical Significance of In-Stent Restenosis in Patients with Diabetes. Int J Environ Res Public Health. 2021 Nov 15;18(22):11970.
Thiruvoipati T, Kielhorn CE, Armstrong EJ. Peripheral artery disease in patients with diabetes: Epidemiology, mechanisms, and outcomes. World J Diabetes. 2015 Jul 10;6(7):961-9.
Comment-2
I do not understand the concept of cDDD. The Authors wrote that cDDD is the sum of DDD for all the PPIs. But one patient takes only one PPI at the moment. So why do you use the sum for all PPIs? It is not clear for me, please explain.
Response to Comment-2:
We are grateful for the very wisdom suggestion to improve this manuscript.
We revised in the section of “2.3 PPIs exposure” in the “Materials and Methods” part as follows:
The defined daily dose (DDD) recommended by the World Health Organization (WHO) is a unit for measuring the prescribed amount of drugs. The DDD is the assumed average maintenance dose per day of a drug consumed for its main indication. We calculated the sum of the dispensed DDD as cumulative DDD (cDDD) of PPI use during the follow-up period. Based on the cDDD, PPI use patterns were separated into four subgroups: cDDD ≤ 30, cDDD 31–120, cDDD 121–365, and cDDD > 365. By using cDDD, we enable comparison of drug usage between different drugs in the same group and allow to examine the dose-response effect of PPIs on T2DM, which is commonly used in previous pharmacoepidemiological studies of PPIs [27-29].
Comment-3
In the Discussion, when you cite the results obtained by other researchers and use hazard or odds ratio, give also always confidence interval, please (for example line 207).
Response to Comment-3:
Thank you for the kind reminding. We revised it accordingly.
A recent prospective cohort study conducted by Yuan et al. recruited 204,689 health professionals showed that regular PPIs use was associated with a 24% higher risk of T2DM (hazard ratio [HR] 1.24, 95% CI: 1.17-1.31), and the risk increased with a longer duration of PPIs exposure [11]. However, this study was based on a certain population rather than the general population and did not provide detailed information regarding the dosage, brand, frequency, or indication of PPIs use. Another prospective cohort study conducted with 9,531 participants showed that incident PPI use was associated with a significantly increased risk of T2DM, with a hazard of 1.69 (95% CI: 1.36-2.10) [10]. In addition, they found that the risk of T2DM increased with a higher dosage and longer duration of PPIs exposure, which is consistent with our findings.
On the other hand, other studies have shown contrary results to our study. For example, a retrospective cohort study by Lin et al. based on the NHIRD included 388,098 patients, which showed that patients with upper gastrointestinal disease (UGID) receiving PPIs had a20% decreased risk of diabetes compared with patients with UGID without PPIs use over a 5-year-follow-upperiod (HR: 0.80, 95% CI: 0.73-0.88) [21].
Comment-4
In the Discussion the Authors used such expression as low-potency and high-potency PPIs. Please, explain precisely which substances are low-patency PPIs and which ones are high-patency.
Response to Comment-4:
Thank you for the kind reminding.
There is no absolute classification of high- and low- potency of PPIs, except for relatively potency. Based on Kirchheiner et al. reported (citation reference 30 in our original manuscript), the relative potencies according to the mean 24-h gastric pH of the five PPIs compared to omeprazole were 0.23, 0.90, 1.00, 1.60, and 1.82 for pantoprazole, lansoprazole, omeprazole, esomeprazole, and rabeprazole, respectively. To echo previous study, we used “relatively” high and “relatively” low potency as follow:
The potency of the five PPIs, from low to high, was pantoprazole, lansoprazole, omeprazole, esomeprazole, and rabeprazole [30]. Our findings show that PPI with lower potency seems to have a higher risk of T2DM. Patients taking PPIs with relatively lower potency, such as pantoprazole, may need higher equivalent dosage to reach the anti-acid effect as those with relatively higher potency. For example, 20mg of esomeprazole is approximately equivalent to 120mg of Pantoprazole [31]. Higher dosage exposure to PPIs with relatively lower potency may partially explain the increased risk of T2DM. Further studies investigating the underlying mechanism of PPIs and their metabolic effects are needed.
New reference
Graham, D. Y.; Lu, H.; Dore, M. P., Relative potency of proton-pump inhibitors, Helicobacter pylori therapy cure rates, and meaning of double-dose PPI. Helicobacter 2019, 24, e12554.
Comment-5
Please, comment how dosage intervals used in the analysis are related to typical doses of PPIs used in the clinical practice.
Response to Comment-5:
Thank you for the kind reminding. We revised it accordingly as follows:
We calculated the sum of the dispensed DDD as cumulative DDD (cDDD) of PPI use during the follow-up period. Based on the cDDD, PPI use patterns were separated into four subgroups: cDDD ≤ 30, cDDD 31–120, cDDD 121–365, and cDDD > 365. Such dosage intervals were based on methodology of previous studies [Li et al., 2021; Liang et al., 2021] Patients in the cDDD > 365 group may represent a person who had continuously received PPI treatment on standard dosage for one year. By using cDDD, we enable comparison of drug usage between different drugs in the same group and allow to examine the dose-response effect of PPIs on T2DM, which is commonly used in previous pharmacoepidemiological studies of PPIs.
New reference
Li CY, Dai YX, Chang YT, Bai YM, Tsai SJ, Chen TJ, Chen MH. Proton Pump Inhibitors Are Associated with Increased Risk of Psoriasis: A Nationwide Nested Case-Control Study. Dermatology. 2021;237(6):884-890.
Liang CS, Bai YM, Hsu JW, Huang KL, Ko NY, Tsai CK, Yeh TC, Chu HT, Tsai SJ, Chen TJ, Chen MH. The Risk of Epilepsy after Long-term Proton Pump Inhibitor Therapy. Seizure. 2021 Apr;87:88-93.
Minor Comment-1
In the main text references should be marked in square brackets, e.g. [1], [2,3], etc.
Response to Minor Comment-1:
Thank you for the kind reminding. We revised it accordingly across the whole manuscript.
Minor Comment-2
Please check the work very carefully for minor editing errors, such as the lack of spaces between some words or sentences, as in lines 71, 74, 96, 200, 205.
Response to Minor Comment-2:
Thank you for the kind reminding and checking. We revised it accordingly.
Minor Comment-3
List of references should be written in accordance with the rules of MDPI.
Response to Minor Comment-3:
Thank you for the kind reminding. We revised it accordingly.

Reviewer 2 Report
This study was addressed to obtain significant evidences able to settle the “vexata quaestio” (debated and controversial question) concerning the real ability of treatment with proton pump inhibitors (PPIs) to increase the risk of type 2 diabetes mellitus (T2DM). Although a long-term use of PPI has been reported to induce adverse effects such as gastrointestinal infections and bone diseases, some previous studies on PPIs and increased risk of T2DM have provided negative conclusions about this association or have shown findings not applicable to the general population or have not considered variables like PPIs dosage, treatment frequency, brand and clinical indications or have not well matched individual PPIs with follow-up periods.
The present case-control study was carried out by getting informations from Taiwan’s National Health Insurance Research Database (1998-2013) with inclusion of 20,940 subjects with T2DM and 20,940 controls. The dosages of PPIs were assessed according to their cumulative defined daily doses (DDD). It was found that higher doses of PPIs were associated with a major risk of T2DM development for which pantoprazole, lansoprazole and omeprazole appeared to play a prevailing role. Moreover, such higher doses increased the above risk dose-dependently among patients affected by upper gastrointestinal diseases.
In Materials and Methods, Authors adequately indicated the inclusion criteria that were applied to select controls and subjects with T2DM. Also DDD was duly defined and confounding factors (such as obesity, dyslipidemia and alcohol use) were considered. Statistical analysis was performed through a series of parametric and non-parametric tests including the independent t-test, the Pearson’s X2 test and the conditional logistic regression analysis (able to ascertain the likelihood of T2DM after assessment of variables such as demographic data, indications of PPIs employment and concomitant comorbidities). Also sex and categories of PPIs were adequately considered.
Results were correctly reported with particular emphasis on associations linking PPIs exposure and individual PPIs with risk of T2DM. Discussion resumes the main evidences of the study, i.e. the increased risk of T2DM following rising dosages of PPIs without differences between male and female patients and between patients with or without medical comorbidities. Interestingly, some PPIs were found to increase the risk of T2DM (i.e. pantoprazole, lansoprazole and omeprazole), whereas other PPIs (i.e. esomeprazole or rabeprazole) did not show such effect. Discussion also dealt with other studies on PPIs and T2DM evidencing, as above stated, how their results have lead to controversial conclusions depending on methodological designs, studied populations, follow-up periods, medical comorbidities and so on. Authors correctly underline how only speculations may be advanced on mechanisms by which PPIs can favour the occurrence of T2DM. In this regard, it has been speculated that PPIs interfere with glucose metabolism, gut microbiota and magnesium homeostasis. Conclusively, Authors highlight strengths and limitations of their study, the latter mainly ascribable to a series of uncontrollable variables concerning the subjects that were recruited (e.g. family history of T2DM, lifestyle and environmental factors, medical adherence to medications, etc.).
Overall, the present study is somewhat interesting in having tried to better define relationships linking the employment of PPIs with a risk of T2DM. A merit of the study is to have operated on a large and well-defined patient sample and to have matched the study groups in relation to a series of confounding factors. However, the relationship between PPIs and T2DM remains to be clarified about the involved pharmacodynamic/pharmacotoxicological mechanisms, which reputedly operate within a multifactorial pathogenesis. Analogously, there is uncertainty about other adverse actions of PPIs such as the reported renal damage, altered absorption of vitamins and minerals, favoured infections, hypergastrinemia-related problems, multifocal atrophic gastritis, intestinal metaplasia and drug interactions ranging from clopidogrel to diazepam, digoxin and antifungal drugs. Conclusively, the present study has been performed with methodological accuracy allowing significant results. The manuscript has been prepared with appreciable care, in which references are updated and appropriate and the four tables are well explicative. Grammar, lexicon, “English style” and sentence fluency are acceptable.
Some minor corrections should be made through the text where, here and there, some sentences have to be readjusted. For example:
- Title, line 2: Dose-Depend…=Dose-Dependent…
- Abstract, lines 29-31: A higher dose of PPI…than that of the lower dose…= A dose-dependent increased risk of T2DM was evidenced, among patients diagnosed with UGID, when using higher doses of PPIs, being such risk greater than that observed with lower doses of these drugs.
- Please space out a series of pooled words as in line 203 (ratherthanthe).
Author Response
#Reviewer 2
This study was addressed to obtain significant evidences able to settle the “vexata quaestio” (debated and controversial question) concerning the real ability of treatment with proton pump inhibitors (PPIs) to increase the risk of type 2 diabetes mellitus (T2DM). Although a long-term use of PPI has been reported to induce adverse effects such as gastrointestinal infections and bone diseases, some previous studies on PPIs and increased risk of T2DM have provided negative conclusions about this association or have shown findings not applicable to the general population or have not considered variables like PPIs dosage, treatment frequency, brand and clinical indications or have not well matched individual PPIs with follow-up periods.
The present case-control study was carried out by getting informations from Taiwan’s National Health Insurance Research Database (1998-2013) with inclusion of 20,940 subjects with T2DM and 20,940 controls. The dosages of PPIs were assessed according to their cumulative defined daily doses (DDD). It was found that higher doses of PPIs were associated with a major risk of T2DM development for which pantoprazole, lansoprazole and omeprazole appeared to play a prevailing role. Moreover, such higher doses increased the above risk dose-dependently among patients affected by upper gastrointestinal diseases.
In Materials and Methods, Authors adequately indicated the inclusion criteria that were applied to select controls and subjects with T2DM. Also DDD was duly defined and confounding factors (such as obesity, dyslipidemia and alcohol use) were considered. Statistical analysis was performed through a series of parametric and non-parametric tests including the independent t-test, the Pearson’s X2 test and the conditional logistic regression analysis (able to ascertain the likelihood of T2DM after assessment of variables such as demographic data, indications of PPIs employment and concomitant comorbidities). Also sex and categories of PPIs were adequately considered.
Results were correctly reported with particular emphasis on associations linking PPIs exposure and individual PPIs with risk of T2DM. Discussion resumes the main evidences of the study, i.e. the increased risk of T2DM following rising dosages of PPIs without differences between male and female patients and between patients with or without medical comorbidities. Interestingly, some PPIs were found to increase the risk of T2DM (i.e. pantoprazole, lansoprazole and omeprazole), whereas other PPIs (i.e. esomeprazole or rabeprazole) did not show such effect. Discussion also dealt with other studies on PPIs and T2DM evidencing, as above stated, how their results have lead to controversial conclusions depending on methodological designs, studied populations, follow-up periods, medical comorbidities and so on. Authors correctly underline how only speculations may be advanced on mechanisms by which PPIs can favour the occurrence of T2DM. In this regard, it has been speculated that PPIs interfere with glucose metabolism, gut microbiota and magnesium homeostasis. Conclusively, Authors highlight strengths and limitations of their study, the latter mainly ascribable to a series of uncontrollable variables concerning the subjects that were recruited (e.g. family history of T2DM, lifestyle and environmental factors, medical adherence to medications, etc.).
Overall, the present study is somewhat interesting in having tried to better define relationships linking the employment of PPIs with a risk of T2DM. A merit of the study is to have operated on a large and well-defined patient sample and to have matched the study groups in relation to a series of confounding factors. However, the relationship between PPIs and T2DM remains to be clarified about the involved pharmacodynamic/pharmacotoxicological mechanisms, which reputedly operate within a multifactorial pathogenesis. Analogously, there is uncertainty about other adverse actions of PPIs such as the reported renal damage, altered absorption of vitamins and minerals, favoured infections, hypergastrinemia-related problems, multifocal atrophic gastritis, intestinal metaplasia and drug interactions ranging from clopidogrel to diazepam, digoxin and antifungal drugs. Conclusively, the present study has been performed with methodological accuracy allowing significant results. The manuscript has been prepared with appreciable care, in which references are updated and appropriate and the four tables are well explicative. Grammar, lexicon, “English style” and sentence fluency are acceptable.
Response to Comment:
We appreciate the reviewer’s time and effort in enhancing the quality of our work. We have amended the manuscript following your suggestions. We hope that we have appropriately addressed all the comments and the manuscript is now suitable for publication.
Comment
Some minor corrections should be made through the text where, here and there, some sentences have to be readjusted. For example:
-Title, line 2: Dose-Depend…=Dose-Dependent…
-Abstract, lines 29-31: A higher dose of PPI…than that of the lower dose…= A dose-dependent increased risk of T2DM was evidenced, among patients diagnosed with UGID, when using higher doses of PPIs, being such risk greater than that observed with lower doses of these drugs.
-Please space out a series of pooled words as in line 203 (ratherthanthe).
Response to Comment:
We thank you for raising these points. We revised it accordingly. Thank you AGAIN.
(1) We revised in the title of dose-depend to “dose-dependent”.
(2) In the conclusion of abstract, we revised as follows: A dose-dependent increased risk of T2DM was found among patients with UGID using higher doses of PPIs compared than those with lower doses of these drugs. Further studies are necessary to investigate the underlying pathophysiology of PPIs and T2DM.
(3) We corrected it as “rather than the”.

Round 2
Reviewer 1 Report
In my opinion, the paper has been significantly improved. The Authors answer my questions in a satisfactory way. I recommend the paper for publication in its current form.